# COMBATING ADVERSARIAL ATTACKS USING SPARSE REPRESENTATIONS

**Soorya Gopalakrishnan**[*]**, Zhinus Marzi**[*]**, Upamanyu Madhow & Ramtin Pedarsani**
Department of Electrical and Computer Engineering
University of California, Santa Barbara
Santa Barbara, CA 93106, USA
{soorya,zhinus_marzi,madhow,ramtin}@ucsb.edu

## ABSTRACT

It is by now well-known that small *adversarial* perturbations can induce classification errors in deep neural networks (DNNs). In this paper, we make the case that sparse representations of the input data are a crucial tool for combating such attacks. For linear classifiers, we show that a sparsifying front end is *provably* effective against $\ell_\infty$-bounded attacks, reducing output distortion due to the attack by a factor of roughly $K/N$ where $N$ is the data dimension and $K$ is the sparsity level. We then extend this concept to DNNs, showing that a "locally linear" model can be used to develop a theoretical foundation for crafting attacks and defenses. Experimental results for the MNIST dataset show the efficacy of the proposed sparsifying front end.

## 1 INTRODUCTION

It has been less than five years since Szegedy et al. (2014) and Goodfellow et al. (2015) pointed out the vulnerability of deep networks to tiny, carefully designed *adversarial* perturbations, but there is now widespread recognition that understanding and combating such attacks is a crucial challenge in machine learning security. It was conjectured by Goodfellow et al. (2015) (see also later work by Moosavi-Dezfooli et al. (2016); Poole et al. (2016); Fawzi et al. (2017)) that the vulnerability arises not because deep networks are complicated and nonlinear, but because they are "too linear." We argue here that this intuition is spot on, using it to develop a systematic, theoretically grounded, framework for design of both attacks and defenses, and making the case that sparse input representations are a critical tool for defending against $\ell_\infty$-bounded perturbations.

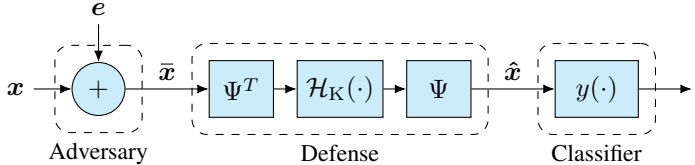

Figure 1: Defense against adversarial attacks via a sparsifying front end

Figure 1 depicts a classifier attacked by an adversary which can corrupt the input $x$ by adding a perturbation $e$ satisfying $\|e\|_\infty \le \epsilon$, and a defense based on a sparsifying front end, which exploits the rather general observation that input data must be sparse in *some* basis in order to avoid the curse of dimensionality. Specifically, we assume that input data $x \in \mathbb{R}^N$ has a $K$-sparse representation ($K \ll N$) in an orthonormal basis $\Psi$: $\left\| \Psi^T x \right\|_0 \le K$. The front end enforces sparsity in domain $\Psi$ via function $\mathcal{H}_K(\cdot)$, retaining the $K$ coefficients largest in magnitude and zeroing out the rest.

The intuition behind why sparsity can help is quite clear: small perturbations can add up to a large output distortion when the input dimension is large, and by projecting to a smaller subspace, we limit the damage. Indeed, many recently proposed defenses implicitly use some notion of sparsity, such as

---

[*]Joint first authors.

JPEG compression (Das et al., 2017; Guo et al., 2018), PCA (Bhagoji et al., 2017), and projection onto GAN-based generative models (Ilyas et al., 2017; Samangouei et al., 2018). Our goal here is to provide a theoretically grounded framework which permits a systematic pursuit of sparsity as a key tool, perhaps *the* key tool, for robust machine learning.

We first motivate our approach by rigorous results for linear classifiers, and then show how the approach extends to general neural networks via a "locally linear" model.

## 2 LINEAR CLASSIFIERS

For a linear classifier, $y(\boldsymbol{x}) = \boldsymbol{w}^T \boldsymbol{x}$ and hence the distortion caused by the adversary is $\Delta = \left| \boldsymbol{w}^T \hat{\boldsymbol{x}} - \boldsymbol{w}^T \boldsymbol{x} \right|$. Denote the support of the $K$-sparse representation of $\boldsymbol{x}$ by $\mathcal{S}_\mathrm{K}(\boldsymbol{x}) = \mathrm{supp}\big(\mathcal{H}_\mathrm{K}\big(\Psi^T \boldsymbol{x}\big)\big)$. We say that we are in a *high signal-to-noise ratio (SNR)* regime when the support does not change due to the perturbation: $\mathcal{S}_\mathrm{K}(\boldsymbol{x}) = \mathcal{S}_\mathrm{K}(\boldsymbol{x} + \boldsymbol{e})$. The high SNR regime can be characterized as follows:

**Proposition 1.** *For sparsity level K, a sufficient condition for high SNR is: $\lambda/\epsilon > 2M$, where $\lambda$ is the magnitude of the smallest non-zero entry of $\mathcal{H}_\mathrm{K}(\Psi^T x)$, and $M = \max_j \|\boldsymbol{\psi}_j\|_1$.*[1]

We note that the high SNR condition is easier to satisfy for bases with sparser, or more localized, basis functions (smaller $M$).

The distortion now becomes $\Delta = \left| \boldsymbol{e}^T \mathcal{P}_\mathrm{K}(\boldsymbol{w}, \boldsymbol{x}) \right|$, where $\mathcal{P}_\mathrm{K}(\boldsymbol{w}, \boldsymbol{x}) = \sum_{k \in \mathcal{S}_\mathrm{K}(\boldsymbol{x})} \boldsymbol{\psi}_k \boldsymbol{\psi}_k^T \boldsymbol{w}$ is the projection of $\boldsymbol{w}$ onto the $K$-dimensional subspace spanned by $\mathcal{S}_\mathrm{K}(\boldsymbol{x})$. We now consider two settings:

- *Semi-white box:* Here the perturbations are designed based on knowledge of $\boldsymbol{w}$ alone, so that $\boldsymbol{e}_\mathrm{SW} = \epsilon \, \mathrm{sgn}(\boldsymbol{w})$ and $\Delta_\mathrm{SW} = \epsilon \left| \mathrm{sgn}(\boldsymbol{w}^T) \, \mathcal{P}_\mathrm{K}(\boldsymbol{w}, \boldsymbol{x}) \right|$. We note that the attack is aligned with $\boldsymbol{w}$.

- *White box:* Here the adversary has knowledge of both $\boldsymbol{w}$ and the front end. Assuming high SNR, the optimal perturbation is $\boldsymbol{e}_\mathrm{W} = \epsilon \, \mathrm{sgn}(\mathcal{P}_\mathrm{K}(\boldsymbol{w}, \boldsymbol{x}))$, which yields $\Delta_\mathrm{W} = \epsilon \|\mathcal{P}_\mathrm{K}(\boldsymbol{w}, \boldsymbol{x})\|_1$. Thus, instead of aligning with $\boldsymbol{w}$, $\boldsymbol{e}_\mathrm{W}$ is aligned to the projection of $\boldsymbol{w}$ on the subspace that $\boldsymbol{x}$ lies in.

In order to understand how well the sparsifying front end works, we take an ensemble average over randomly chosen classifiers $\boldsymbol{w}$, and show (Marzi et al., 2018) that, relative to no defense, the attenuation in output distortion provided by the sparsifying front end is a factor of $K/N$ for a semi-white box attack, and is at least $\mathcal{O}(K \, \mathrm{polylog}(N)/N)$ for the white box attack. We do not state these theorems formally here due to lack of space. A practical take-away from the calculations involved is that, in order for the defense to be effective against a white box attack, not only do we need $K \ll N$, but we also need that the individual basis functions be localized (small in $\ell_1$ norm).

## 3 NEURAL NETWORKS

We skip a lot of details, but the key idea is to extend the intuition from linear classifiers to general neural networks using the concept of a "locally linear" representation. The change in slope in a ReLU function, or the selection of the maximum in a max pooling function, can be modeled as an input-dependent switch. If we fix these switches, the transfer function from the input to the network to, say, the inputs to an output softmax layer, is linear. Specifically, consider a multilayer (deep) network with $L$ classes. Using the locally linear model, each of the outputs of the network (prior to softmax) can be written as

$$y_i = \boldsymbol{w}_\mathrm{eq}^{\{i\}^T} \boldsymbol{x} - b_\mathrm{eq}^{\{i\}}, \quad i = 1, \ldots, L,$$

where $\boldsymbol{y} = [y_1, y_2, ..., y_L]^T$. The softmax layer computes $p_i = S_i(\boldsymbol{y}) = e^{y_i} / \big(\sum_{j=1}^L e^{y_j}\big)$.

Applying the theory developed for linear classifiers to $\boldsymbol{w}_\mathrm{eq}^{\{i\}}$, we see that a sparsifying front end will attenuate the distortion going in to the softmax layer. And of course, the adversary can use the locally linear model to devise attacks analogous with those for linear classifiers, as follows.

*Semi-white box and white box attacks:* Assume that $\boldsymbol{x}$ belongs to class $t$, with label $t$ known to the adversary (a pessimistic but realistic assumption, given that the attacker can run the input through a high-accuracy network prior to devising its attack). The adversary can sidestep the nonlinearity

---

[1] Here $\boldsymbol{\psi}_j$ denotes the $j$th column of $\Psi$, i.e. $\Psi = [\boldsymbol{\psi}_1, \boldsymbol{\psi}_2, \ldots, \boldsymbol{\psi}_N]$.

Table 1: Classification accuracies (in %) for 3 vs. 7 discrimination via linear SVM, and 10-class classification via CNN. For linear SVM, $\epsilon = 0.12$ and $\rho = 2\%$. For the CNN, $\epsilon = 0.25$, $\rho = 3\%$.

| | Linear SVM | | Four layer CNN | | |
|---|---|---|---|---|---|
| | Semi-white box | White box | FGSM | Semi-white box | White box |
| No defense | 0 | 0 | 19.45 | 8.87 | 8.87 |
| Sparsifying front end | 97.31 | 94.62 | 89.75 | 88.76 | 84.04 |

of the softmax layer, since its goal is simply to make $y_i > y_t$ for *some* $i \neq t$. Thus, the adversary can consider $L-1$ binary classification problems, and solve for perturbations aiming to maximize $y_i - y_t$ for each $i \neq t$. We now apply the semi-white and white box attacks to each pair, with $\boldsymbol{w}_{\text{eq}} = \boldsymbol{w}_{\text{eq}}^{\{i\}} - \boldsymbol{w}_{\text{eq}}^{\{t\}}$ being the equivalent locally linear model from the input to $y_i - y_t$. After computing the distortions for each pair, the adversary applies its attack budget to the *worst-case* pair for which the distortion is the largest: $\max_{i,\boldsymbol{e}} y_i(\boldsymbol{x} + \boldsymbol{e}) - y_t(\boldsymbol{x} + \boldsymbol{e})$ s.t. $\|\boldsymbol{e}\|_\infty \leq \epsilon$.

*FGSM attack:* For *binary* classification, the standard FGSM attack (Goodfellow et al., 2015) can be shown to be equivalent to the semi-white box attack using the locally linear model. However, it does not have such a nice interpretation for more than two classes: it attacks along the gradient of the cost function (typically cross-entropy), and hence does not take as direct an approach to confounding the network as the semi-white box attack. As expected (and verified by experiments), it performs worse (does less damage) than the semi-white box attack.

## 4    EXPERIMENTS

**Setup:** We consider two inference tasks on the MNIST dataset (LeCun et al., 1998): binary classification of digit pairs via linear SVM, and multi-class classification via a four layer CNN. The CNN consists of two convolutional layers (containing 20 and 40 feature maps, both with 5x5 local receptive fields) and two fully connected layers (containing 1000 neurons each, with dropout) (Nielsen, 2015). For the sparsifying front end, we use the Cohen-Debauchies-Feauveau 9/7 wavelet (Cohen et al., 1992) and retrain the network with sparsified images for various values of $\rho = K/N$. We perturb images with FGSM, semi-white box and white box attacks and report on classification accuracies.[2]

**Results:** For the binary classification task, we begin with the digits 3 and 7. Without the front end, an attack[3] with $\epsilon = 0.12$ completely overwhelms the classifier, reducing accuracy from 98.2% to 0%. We find $\rho = 2\%$ to be the best choice for the 3 versus 7 scenario, and report on the accuracies obtained in Table 1. Results for other digit pairs show a similar trend; insertion of the front end greatly improves resilience to adversarial attacks. The optimal value of $\rho$ lies between 1-5%, with $\rho = 2\%$ working well for all scenarios.

For multi-class classification, the attacks use $\epsilon = 0.25$. We find that the locally linear attack is stronger than FGSM; it degrades performance from 99.38% to 8.87% when no defense is present. Again, the sparsifying front end ($\rho = 3\%$) improves network robustness, increasing accuracy to 84.04% in the worst-case scenario.

## 5    CONCLUSIONS

We have emphasized here the value of locally linear modeling for design of attacks and defenses, and the connection between sparsity and robustness. We believe that these results just scratch the surface, and hope that they stimulate the community towards developing a comprehensive design framework, grounded in theoretical fundamentals, for robust neural networks. Important topics for future work include developing better sparse generative models, as well as discriminative approaches to sparsity (e.g., via sparsity of weights within the neural network). Promising results on the latter approach have been omitted here due to lack of space.

---

[2] Code is available at `https://github.com/soorya19/sparsity-based-defenses`.

[3] The reported values of $\epsilon$ are for images normalized to $[0, 1]$.

ACKNOWLEDGMENT

This work was supported in part by the National Science Foundation under grants CNS-1518812 and CCF-1755808, by Systems on Nanoscale Information fabriCs (SONIC), one of the six SRC STARnet Centers, sponsored by MARCO and DARPA, and by the UC Office of the President under grant No. LFR-18-548175.

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
