# OpenReview forum: "Combating Adversarial Attacks Using Sparse Representations"
_ICLR.cc/2018/Workshop — Accept_

### Official Review · AnonReviewer2 · 2018-03-09
**Sparse Representations in NNs to Defend Against Adversarial Attacks**

**Rating:** 8
**Confidence:** 2

**Review:**

In this paper, the authors discuss sparse representations as a tool to combat adversarial attacks on neural networks. The idea is that if the input dimension is projected to a smaller subspace, the effects of small perturbations in the input space is lessened. They use a theoretical framework to prove guarantees about their approach, and apply their methods to MNIST to show that they are better equipped to deal with adversarial attacks than non-sparse models.

This was a well-written paper with an interesting idea, and the authors provide a lot of intuition. My main comment is that it skips a lot of background -- I would suggest adding more background about adversarial attacks and the terminology used.

---

### Official Review · AnonReviewer3 · 2018-03-09
**interesting idea; a little terse**

**Rating:** 6
**Confidence:** 3

**Review:**

the paper presents an interesting extension from sparsifying front end for linear classifiers to deep neural networks. this is potentially an interesting idea. I believe the paper could benefit from expanding sec2: the proof of sparsifying front end effective against l_\infty-bounded attacks; the current terseness has hindered my understanding of the precise argument. (the intuition makes sense though.)

---

### Official Review · AnonReviewer1 · 2018-03-09
**Interesting work on use of a sparsifying front end to linear/neural network models to tackle adversarial attacks with promising results.**

**Rating:** 6
**Confidence:** 3

**Review:**

The submitted work, aims to present theoretical bases to demonstrate the efficacy of sparsifying input data to linear/neural network models in order to tackle adversarial attacks.
The aim of the work is to propose a "systematic, theoretically grounded, framework for design of both attacks and defenses". A short workshop paper like this one does not provide all elements to judge such statement but the approach is clear.
The application of the approach to a linear model and then expanding the evaluation to a neural network via a "locally" linear model is well laid.
Application of the approach to varying dimensionality spaces and determination of the corresponding best sparsity values to handle most adversarial attacks would be interesting to analyze.
The results on the MNIST handwritten digit dataset, both for SVM and a four layer CNN look promising.

---

### Decision · Program_Chairs · 2018-03-20
**ICLR 2018 Workshop Acceptance Decision**

**Decision:**

Accept

**Comment:**

Congratulations, your paper was accepted to the ICLR workshop.